# Step Size Matters in Deep Learning

**Kamil Nar**      **S. Shankar Sastry**
Electrical Engineering and Computer Sciences
University of California, Berkeley
{nar,sastry}@eecs.berkeley.edu

## Abstract

Training a neural network with the gradient descent algorithm gives rise to a discrete-time nonlinear dynamical system. Consequently, behaviors that are typically observed in these systems emerge during training, such as convergence to an orbit but not to a fixed point or dependence of convergence on the initialization. Step size of the algorithm plays a critical role in these behaviors: it determines the subset of the local optima that the algorithm can converge to, and it specifies the magnitude of the oscillations if the algorithm converges to an orbit. To elucidate the effects of the step size on training of neural networks, we study the gradient descent algorithm as a discrete-time dynamical system, and by analyzing the Lyapunov stability of different solutions, we show the relationship between the step size of the algorithm and the solutions that can be obtained with this algorithm. The results provide an explanation for several phenomena observed in practice, including the deterioration in the training error with increased depth, the hardness of estimating linear mappings with large singular values, and the distinct performance of deep residual networks.

## 1   Introduction

When gradient descent algorithm is used to minimize a function, say $f : \mathbb{R}^n \to \mathbb{R}$, it leads to a discrete-time dynamical system:

$$x[k+1] = x[k] - \delta \nabla f(x[k]), \tag{1}$$

where $x[k]$ is *the state* of the system, which consists of the parameters updated by the algorithm, and $\delta$ is the step size, or the learning rate of the algorithm. Every fixed point of the system (1) is called *an equilibrium* of the system, and they correspond to the critical points of the function $f$.

Unless $f$ is a quadratic function of the parameters, the system described by (1) is either a nonlinear system or a hybrid system that switches from one dynamics to another over time. Consequently, the system (1) can exhibit behaviors that are typically observed in nonlinear and hybrid systems, such as convergence to an orbit but not to a fixed point, or dependence of convergence on the equilibria and the initialization. The step size of the algorithm has a critical effect on these behaviors, as shown in the following examples.

**Example 1. Convergence to a periodic orbit:** Consider the continuously differentiable and convex function $f_1(x) = \frac{2}{3}|x|^{3/2}$, which has a unique local minimum at the origin. The gradient descent algorithm on this function yields

$$x[k+1] = \begin{cases} x[k] - \delta\sqrt{x[k]}, & x[k] \geq 0, \\ x[k] + \delta\sqrt{-x[k]}, & x[k] < 0. \end{cases}$$

As expected, the origin is the only equilibrium of this system. Interestingly, however, $x[k]$ converges to the origin only when the initial state $x[0]$ belongs to a countable set $\mathcal{S}$:

$$\mathcal{S} = \left\{ 0, \delta^2, -\delta^2, \frac{3+\sqrt{5}}{2}\delta^2, -\frac{3+\sqrt{5}}{2}\delta^2, \dots \right\}.$$

For all other initializations, $x[k]$ converges to an oscillation between $\delta^2/4$ and $-\delta^2/4$. This implies that, if the initial state $x[0]$ is randomly drawn from a continuous distribution, then almost surely $x[k]$ does not converge to the origin, yet $|x[k]|$ converges to $\delta^2/4$. In other words, with probability 1, the state $x[k]$ does not converge to a fixed point, such as a local optimum or a saddle point, even though the estimation error converges to a finite non-optimal value.

**Example 2. Dependence of convergence on the equilibrium:** Consider the nonconvex function $f_2(x) = (x^2 + 1)(x - 1)^2(x - 2)^2$, which has two local minima at $x = 1$ and $x = 2$ as shown in Figure 1. Note that these local minima are also the two of the isolated equilibria of the dynamical system created by the gradient descent algorithm. The *stability* of these equilibria *in the sense of Lyapunov* is determined by the step size of the algorithm. In particular, since the *smoothness* parameter of $f_2$ around these equilibria is 4 and 10, they are stable only if the step size is smaller than 0.5 and 0.2, respectively, and the gradient descent algorithm can converge to them only when these conditions are satisfied. Due to the difference in the largest step size allowed for different equilibria, step size conveys information about the solution that can be obtained by the gradient descent algorithm. For example, if the algorithm converges to an equilibrium with step size 0.3 from a point drawn randomly from a continuous distribution, then this equilibrium is almost surely $x = 1$.

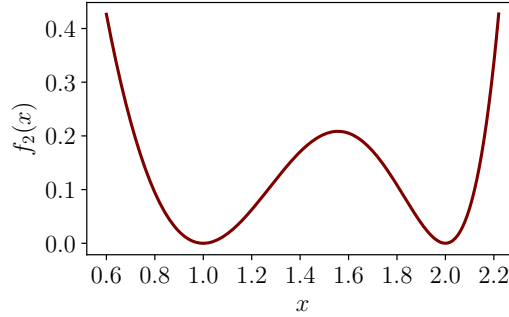

Figure 1: The function $f_2(x) = (x^2 + 1)(x - 1)^2(x - 2)^2$ of Example 2. Since the smoothness parameter of $f_2$ at $x = 1$ is smaller than that at $x = 2$, the gradient descent algorithm cannot converge to $x = 2$ but can converge to $x = 1$ for some values of the step size. If, for example, the algorithm converges to an equilibrium from a randomly chosen initial point with step size 0.3, then this equilibrium is almost surely $x = 1$.

**Example 3. Dependence of convergence on the initialization:** Consider the function $f_3(x) = x^L$ where $L \in \mathbb{N}$ is an even number larger than 2. The gradient descent results in the system

$$x[k+1] = x[k] - \delta L x[k]^{L-1}.$$

The state $x[k]$ converges to the origin if the initial state satisfies $x[0]^{L-2} < (2/L\delta)$ and $x[k]$ diverges if $x[0]^{L-2} > (2/L\delta)$.

These three examples demonstrate:

1. the convergence of training error does not imply the convergence of the algorithm to a local optimum or a saddle point,

2. the step size determines the magnitude of the oscillations if the algorithm converges to an orbit but not to a fixed point,

3. the step size restricts the set of local optima that the algorithm can converge to,

4. the step size influences the convergence of the algorithm differently for each initialization.

Note that **these are direct consequences of the nonlinear dynamics of the gradient descent algorithm** and not of the (non)convexity of the function to be minimized. While both of the functions in Example 1 and Example 3 are convex, the identical behaviors are observed during the minimization of nonconvex training cost functions of neural networks as well.

## 1.1 Our contributions

In this paper, we study the gradient descent algorithm as a discrete-time dynamical system during training deep neural networks, and we show the relationship between the step size of the algorithm and the solutions that can be obtained with this algorithm. In particular, we achieve the following:

1. We analyze the Lyapunov stability of the gradient descent algorithm on deep linear networks and find different upper bounds on the step size that enable convergence to each solution. We show that for every step size, the algorithm can converge to only a subset of the local optima, and there are always some local optima that the algorithm cannot converge to independent of the initialization.

2. We establish that for deep linear networks, there is a direct connection between the smoothness parameter of the **training loss function** and the largest singular value of the **estimated linear function**. In particular, we show that if the gradient descent algorithm can converge to a solution with a large step size, the function estimated by the network must have small singular values, and hence, the estimated function must have a small Lipschitz constant.

3. We show that symmetric positive definite matrices can be estimated with a deep linear network by initializing the weight matrices as the identity, and this initialization allows the use of the largest step size. Conversely, the algorithm is most likely to converge for an arbitrarily chosen step size if the weight matrices are initialized as the identity.

4. We show that symmetric matrices with negative eigenvalues, on the other hand, cannot be estimated with the identity initialization, and the gradient descent algorithm converges to the closest positive semidefinite matrix in Frobenius norm.

5. For 2-layer neural networks with ReLU activations, we obtain an explicit relationship between the step size of the gradient descent algorithm and the output of the solution that the algorithm can converge to.

## 1.2 Related work

It is a well-known problem that the gradient of the training cost function can become disproportionate for different parameters when training a neural network. Several works in the literature tried to address this problem. For example, changing the geometry of optimization was proposed in (Neyshabur et al., 2017) and a regularized descent algorithm was proposed to prevent the gradients from exploding and vanishing during training.

Deep residual networks, which is a specific class of neural networks, yielded exceptional results in practice with their peculiar structure (He et al., 2016). By keeping each layer of the network close to the identity function, these networks were able to attain lower training and test errors as the depth of the network was increased. To explain their distinct behavior, the training cost function of their linear versions was shown to possess some crucial properties (Hardt & Ma, 2016). Later, equivalent results were also derived for nonlinear residual networks under certain conditions (Bartlett et al., 2018a).

The effect of the step size on training neural networks was empirically investigated in (Daniel et al., 2016). A step size adaptation scheme was proposed in (Rolinek & Martius, 2018) for the stochastic gradient method and shown to outperform the training with a constant step size. Similarly, some heuristic methods with variable step size were introduced and tested empirically in (Magoulas et al., 1997) and (Jacobs, 1988).

Two-layer linear networks were first studied in (Baldi & Hornik, 1989). The analysis was extended to deep linear networks in (Kawaguchi, 2014), and it was shown that all local optima of these networks were also the global optima. It was discovered in (Hardt & Ma, 2016) that the only critical points of these networks were actually the global optima as long as all layers remained close to the identity function during training. The dynamics of training these networks were also analyzed in (Saxe et al., 2013) and (Gunasekar et al., 2017) by assuming an infinitesimal step size and using a continuous-time approximation to the dynamics.

Lyapunov analysis from the dynamical system theory (Khalil, 2002; Sastry, 1999), which is the main tool for our results in this work, was used in the past to understand and improve the training of neural networks – especially that of the recurrent neural networks (Michel et al., 1988; Matsuoka, 1992; Barabanov & Prokhorov, 2002). State-of-the-art feedforward networks, however, have not been analyzed from this perspective.

We summarize the **major differences** between our contributions and the previous works as follows:

1. We relate the vanishing and exploding gradients that arise during training **feedforward** networks to the Lyapunov stability of the gradient descent algorithm.

2. Unlike the continuous-time analyses given in (Saxe et al., 2013) and (Gunesekar et al., 2017), we study the discrete-time dynamics of the gradient descent with an emphasis on the step size. By doing so, we obtain upper bounds on the step size to be used, and we show that the step size restricts the set of local optima that the algorithm can converge to. Note that these results cannot be obtained with a continuous-time approximation.

3. For deep linear networks with residual structure, (Hardt & Ma, 2016) shows that the gradient of the cost function cannot vanish away from a global optimum. This is not enough, however, to suggest the fast convergence of the algorithm. Given a fixed step size, the algorithm may also converge to an oscillation around a local optimum, as in the case of Example 1. We rule out this possibility and provide a step size so that the algorithm converges to a global optimum with a linear rate.

4. We recently found out that the convergence of the gradient descent algorithm was also studied in (Bartlett et al., 2018b) for symmetric positive definite matrices independently of and concurrently with our preliminary work (Nar & Sastry, 2018). However, unlike (Bartlett et al., 2018b), we give an explicit step size value for the algorithm to converge with a linear rate, and we emphasize the fact that the identity initialization allows convergence with the largest step size.

## 2 Upper bounds on the step size for training deep linear networks

Deep linear networks are a special class of neural networks that do not contain nonlinear activations. They represent a linear mapping and can be described by a multiplication of a set of matrices, namely, $W_L \cdots W_1$, where $W_i \in \mathbb{R}^{n_i \times n_{i-1}}$ for each $i \in [L] := \{1, 2, \ldots, L\}$. Due to the multiplication of different parameters, their training cost is never a quadratic function of the parameters, and therefore, the dynamics of the gradient descent algorithm is always nonlinear during training of these networks. For this reason, they provide a simple model to study some of the nonlinear behaviors observed during training neural networks.

Given a cost function $\ell(W_L \cdots W_1)$, if point $\{\hat{W}_i\}_{i \in [L]}$ is a local minimum, then $\{\alpha_i \hat{W}_i\}_{i \in [L]}$ is also a local minimum for every set of scalars $\{\alpha_i\}_{i \in [L]}$ that satisfy $\alpha_1 \alpha_2 \cdots \alpha_L = 1$. Consequently, independent of the specific choice of $\ell$, the training cost function have infinitely many local optima, none of these local optima is isolated in the parameter space, and the cost function is not strongly convex at any point in the parameter space.

Although multiple local optima attain the same training cost for deep linear networks, the dynamics of the gradient descent algorithm exhibits distinct behaviors around these points. In particular, the step size required to render each of these local optima *stable in the sense of Lyapunov* is very different. Since the Lyapunov stability of a point is a necessary condition for the convergence of the algorithm to that point, the step size that allows convergence to each solution is also different, which is formalized in Theorem 1.

**Theorem 1.** *Given a nonzero matrix $R \in \mathbb{R}^{n_L \times n_0}$ and a set of points $\{x_i\}_{i \in [N]}$ in $\mathbb{R}^{n_0}$ that satisfy $\frac{1}{N} \sum_{i=1}^{N} x_i x_i^\top = I$, assume that $R$ is estimated as a multiplication of the matrices $\{W_j\}_{j \in [L]}$ by minimizing the squared error loss*

$$\frac{1}{2N} \sum_{i=1}^{N} \| R x_i - W_L W_{L-1} \ldots W_2 W_1 x_i \|_2^2 \tag{2}$$

where $W_j \in \mathbb{R}^{n_j \times n_{j-1}}$ for all $j \in [L]$. *Then the gradient descent algorithm with random[1] initializa-tion can converge to a solution* $\{\hat{W}_j\}_{j \in [L]}$ *only if the step size* $\delta$ *satisfies*

$$\delta \leq \frac{2}{\sum_{j=1}^{L} p_{j-1}^2 q_{j+1}^2} \tag{3}$$

*where*

$$p_j = \left\| \hat{W}_j \cdots \hat{W}_2 \hat{W}_1 v \right\|, \quad q_j = \left\| u^\top \hat{W}_L \hat{W}_{L-1} \cdots \hat{W}_j \right\| \quad \forall j \in [L],$$

*and* $u$ *and* $v$ *are the left and right singular vectors of* $\hat{R} = \hat{W}_L \cdots \hat{W}_1$ *corresponding to its largest singular value.*

Considering all the solutions $\{\alpha_i \hat{W}_i\}_{i \in [L]}$ that satisfy $\alpha_1 \alpha_2 \cdots \alpha_L = 1$, the bound in (3) can be arbitrarily small for some of the local optima. Therefore, given a fixed step size $\delta$, the gradient descent algorithm can converge to only a subset of the local optima, and there are always some solutions that the algorithm cannot converge to independent of the initialization.

**Remark 1.** Theorem 1 provides a necessary condition for convergence to a specific solution. It rules out the possibility of converging to a large subset of the local optima; however, it does **not** state that given a step size $\delta$, the algorithm converges to a solution which satisfies (3). It might be the case, for example, that the algorithm converges to an oscillation around a local optimum which violates (3) even though there are some other local optima which satisfy (3).

As a necessary condition for the convergence to a global optimum, we can also find an upper bound on the step size independent of the weight matrices of the solution, which is given next.

**Corollary 1.** *For the minimization problem in Theorem 1, the gradient descent algorithm with random initialization can converge to a global optimum only if the step size* $\delta$ *satisfies*

$$\delta \leq \frac{2}{L\rho(R)^{2(L-1)/L}}, \tag{4}$$

*where* $\rho(R)$ *is the largest singular value of* $R$.

**Remark 2.** Corollary 1 shows that, unlike the optimization of the ordinary least squares problem, the step size required for the convergence of the algorithm depends on the parameter to be estimated, $R$. Consequently, estimating linear mappings with larger singular values requires the use of a smaller step size. Conversely, the step size conveys information about the solution obtained if the algorithm converges. That is, if the algorithm has converged with a large step size, then the Lipschitz constant of the function estimated must be small.

**Corollary 2.** *Assume that the gradient descent algorithm with random initialization has converged to a local optimum* $\hat{R} = \hat{W}_L \ldots \hat{W}_1$ *for the minimization problem in Theorem 1. Then the largest singular value of* $\hat{R}$ *almost surely satisfies*

$$\rho(\hat{R}) \leq \left( \frac{2}{L\delta} \right)^{L/(2L-2)}.$$

The smoothness parameter of the training cost function is directly related to the largest step size that can be used, and consequently, to the Lyapunov stability of the gradient descent algorithm. The denominators of the upper bounds (3) and (4) in Theorem 1 and Corollary 1 necessarily provide a lower bound for the smoothness parameter of the training cost function around corresponding local optima. As a result, Theorem 1 implies that there is no finite Lipschitz constant for the gradient of the training cost function over the whole parameter space.

## 3 Identity initialization allows the largest step size for estimating symmetric positive definite matrices

Corollary 1 provides only a necessary condition for the convergence of the gradient descent algorithm, and the bound (4) is not tight for every estimation problem. However, if the matrix to be estimated is symmetric and positive definite, the algorithm can converge to a solution with step sizes close to (4), which requires a specific initialization of the weight parameters.

**Theorem 2.** *Assume that $R \in \mathbb{R}^{n \times n}$ is a symmetric positive semidefinite matrix, and given a set of points $\{x_i\}_{i \in [N]}$ which satisfy $\frac{1}{N} \sum_{i=1}^{N} x_i x_i^\top = I$, the matrix $R$ is estimated as a multiplication of the square matrices $\{W_j\}_{j \in [L]}$ by minimizing*

$$\frac{1}{2N} \sum_{i=1}^{N} \|R x_i - W_L \dots W_1 x_i\|_2^2.$$

*If the weight parameters are initialized as $W_i[0] = I$ for all $i \in [L]$ and the step size satisfies*

$$\delta \le \min \left\{ \frac{1}{L}, \frac{1}{L\rho(R)^{2(L-1)/L}} \right\},$$

*then each $W_i$ converges to $R^{1/L}$ with a linear rate.*

**Remark 3.** Theorem 2 shows that the algorithm converges to a global optimum despite the nonconvexity of the optimization, and it provides a case where the bound (4) is almost tight. The tightness of the bound implies that for the same step size, most of the other global optima are *unstable in the sense of Lyapunov*, and therefore, the algorithm cannot converge to them independent of the initialization. Consequently, using identity initialization allows convergence to a solution which is most likely to be *stable* for an arbitrarily chosen step size.

**Remark 4.** Given that the identity initialization on deep linear networks is equivalent to the zero initialization of linear residual networks (Hardt & Ma, 2016), Theorem 2 provides an alternative explanation for the exceptional performance of deep residual networks as well (He et al., 2016).

When the matrix to be estimated is symmetric but not positive semidefinite, the bound (4) is still tight for some of the global optima. In this case, however, the eigenvalues of the estimate cannot attain negative values if the weight matrices are initialized with the identity.

**Theorem 3.** *Let $R \in \mathbb{R}^{n \times n}$ in Theorem 2 be a symmetric matrix such that the minimum eigenvalue of $R$, $\lambda_{\min}(R)$, is negative. If the weight parameters are initialized as $W_i[0] = I$ for all $i \in [L]$ and the step size satisfies*

$$\delta \le \min \left\{ \frac{1}{1 - \lambda_{\min}(R)}, \frac{1}{L}, \frac{1}{L\rho(R)^{2(L-1)/L}} \right\},$$

*then the estimate $\hat{R} = \hat{W}_L \cdots \hat{W}_1$ converges to the closest positive semidefinite matrix to $R$ in Frobenius norm.*

From the analysis of symmetric matrices, we observe that the step size required for convergence to a global optimum is largest when the singular vector of $R$ corresponding to its largest singular value is amplified or attenuated equally at each layer of the network. If the initial weight matrices affect this vector in the opposite ways, i.e., if some of the layers attenuate this vector and the others amplify it, then the required step size for convergence could be very small.

## 4 Effect of step size on training two-layer networks with ReLU activations

In Section 2, we analyzed the relationship between the step size of the gradient descent algorithm and the solutions that can be obtained by training deep linear networks. A similar relationship exists for nonlinear networks as well. The following theorem, for example, provides an upper bound on the step size for the convergence of the algorithm when the network has two layers and ReLU activations.

**Theorem 4.** *Given a set of points $\{x_i\}_{i \in [N]}$ in $\mathbb{R}^n$, let a function $f : \mathbb{R}^n \to \mathbb{R}^m$ be estimated by a two-layer neural network with ReLU activations by minimizing the squared error loss:*

$$\min_{W,V} \frac{1}{2} \sum_{i=1}^{N} \|W g(V x_i - b) - f(x_i)\|_2^2,$$

*where $g(\cdot)$ is the ReLU function, $b \in \mathbb{R}^r$ is the fixed bias vector, and the optimization is only over the weight parameters $W \in \mathbb{R}^{m \times r}$ and $V \in \mathbb{R}^{r \times n}$. If the gradient descent algorithm with random initialization converges to a solution $(\hat{W}, \hat{V})$, then the estimate $\hat{f}(x) = \hat{W} g(\hat{V} x - b)$ satisfies*

$$\max_{i \in [N]} \|x_i\|_2 \|\hat{f}(x_i)\|_2 \le \frac{1}{\delta}$$

*almost surely.*

Theorem 4 shows that if the algorithm is able to converge with a large step size, then the estimate $\hat{f}(x)$ must have a small magnitude for large values of $\|x\|$.

Similar to Corollary 1, the bound given by Theorem 4 is not necessarily tight. Nevertheless, it highlights the effect of the step size on the convergence of the algorithm. To demonstrate that small changes in the step size could lead to significantly different solutions, we generated a piecewise continuous function $f : [0,1] \to \mathbb{R}$ and estimated it with a two-layer network by minimizing

$$\sum_{i=1}^{N} |Wg(Vx_i - b) - f(x_i)|^2$$

with two different step sizes $\delta \in \{2 \cdot 10^{-4}, 3 \cdot 10^{-4}\}$, where $W \in \mathbb{R}^{1 \times 20}, V \in \mathbb{R}^{20}, b \in \mathbb{R}^{20}$, $N = 1000$ and $x_i = i/N$ for all $i \in [N]$. The initial values of $W, V$ and the constant vector $b$ were all drawn from independent standard normal distributions; and the vector $b$ was kept the same for both of the step sizes used. As shown in Figure 2, training with $\delta = 2 \cdot 10^{-4}$ converged to a fixed solution, which provided an estimate $\hat{f}$ close the original function $f$. In contrast, training with $\delta = 3 \cdot 10^{-4}$ converged to an oscillation and not to a fixed point. That is, after sufficient training, the estimate kept switching between $\hat{f}_{\text{odd}}$ and $\hat{f}_{\text{even}}$ at each iteration of the gradient descent algorithm.[2]

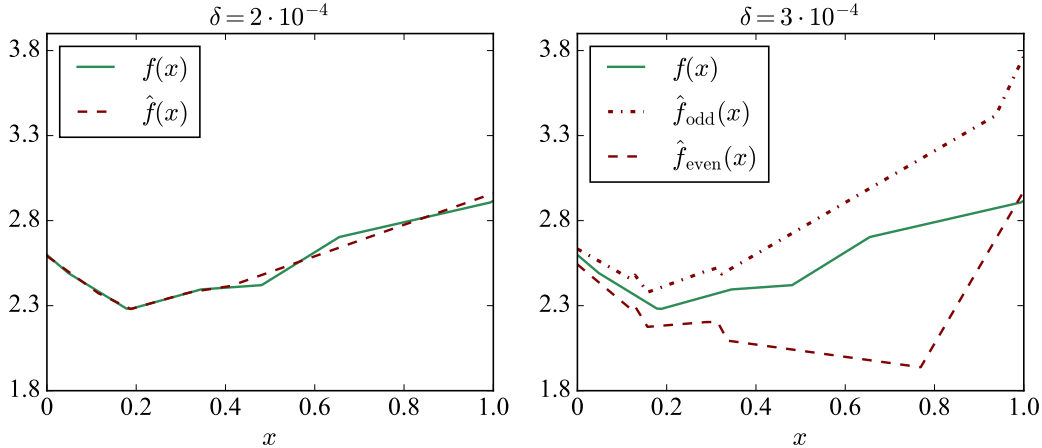

Figure 2: Estimates of the function $f$ obtained by training a two-layer neural network with two different step sizes. **[Left]** When the step size of the gradient descent algorithm is $\delta = 2 \cdot 10^{-4}$, the algorithm converges to a fixed point, which provides an estimate $\hat{f}$ close to $f$. **[Right]** When the step size is $\delta = 3 \cdot 10^{-4}$, the algorithm converges to an oscillation and not to a fixed solution. That is, after sufficient training, the estimate keeps switching between $\hat{f}_{\text{odd}}$ and $\hat{f}_{\text{even}}$ at each iteration.

## 5   Discussion

When gradient descent algorithm is used to minimize a function, typically only three possibilities are considered: convergence to a local optimum, to a global optimum, or to a saddle point. In this work, we considered the fourth possibility: the algorithm may not converge at all – even in the deterministic setting. The training error may not reflect the oscillations in the dynamics, or when a stochastic optimization method is used, the oscillations in the training error might be wrongly attributed to the stochasticity of the algorithm. We underlined that, if the training error of an algorithm converges to a non-optimal value, that does not imply the algorithm is stuck near a bad local optimum or a saddle point; it might simply be the case that the algorithm has not converged at all.

We showed that the step size of the gradient descent algorithm influences the dynamics of the algorithm substantially. It renders some of the local optima unstable in the sense of Lyapunov, and the algorithm cannot converge to these points independent of the initialization. It also determines the magnitude of the oscillations if the algorithm converges to an orbit around an equilibrium point in the parameter space.

In Corollary 2 and Theorem 4, we showed that the step size required for convergence to a specific solution depends on the solution itself. In particular, we showed that there is a direct connection between the smoothness parameter of the training loss function and the Lipschitz constant of the function estimated by the network. This reveals that some solutions, such as linear functions with large singular values, are harder to converge to. Given that there exists a relationship between the Lipschitz constants of the estimated functions and their generalization error (Bartlett et al., 2017), this result could provide a better understanding of the generalization of deep neural networks.

The analysis in this paper was limited to the full-batch gradient descent algorithm. It remains as an open problem to investigate if there are analogous results for the stochastic gradient methods.

## A    Proof of Theorem 1

**Lemma 1.** *Let $f : \mathbb{R}^{m \times n} \to \mathbb{R}^{m \times n}$ be a linear map defined as $f(X) = \sum_{i=1}^{L} A_i X B_i$, where $A_i \in \mathbb{R}^{m \times m}$ and $B_i \in \mathbb{R}^{n \times n}$ are symmetric positive semidefinite matrices for all $i \in [L]$. Then, for every nonzero $u \in \mathbb{R}^m$ and $v \in \mathbb{R}^n$, the largest eigenvalue of $f$ satisfies*

$$\lambda_{\max}(f) \geq \frac{1}{\|u\|_2^2 \|v\|_2^2} \sum_{i=1}^{L} (u^\top A_i u)(v^\top B_i v).$$

**Proof of Theorem 1.** The cost function (2) can be written as

$$\frac{1}{2}\text{trace}\left\{ (W_L \cdots W_1 - R)^\top (W_L \cdots W_1 - R) \right\}.$$

Let $E$ denote the error in the estimate, i.e. $E = W_L \cdots W_1 - R$. The gradient descent yields

$$W_i[k+1] = W_i[k] - \delta W_{i+1}^\top[k] \cdots W_L^\top[k] E[k] W_1^\top[k] \cdots W_{i-1}^\top[k] \quad \forall i \in [L]. \tag{5}$$

By multiplying the update equations of $W_i[k]$ and subtracting $R$, we can obtain the dynamics of $E$ as

$$E[k+1] = E[k] - \delta \sum_{i=1}^{L} A_i[k] E[k] B_i[k] + o(E[k]), \tag{6}$$

where $o(\cdot)$ denotes the higher order terms, and

$$A_i = W_L W_{L-1} \cdots W_{i+1} W_{i+1}^\top \cdots W_{L-1}^\top W_L^\top \quad \forall i \in [L],$$

$$B_i = W_1^\top W_2^\top \cdots W_{i-1}^\top W_{i-1} \cdots W_2 W_1 \quad \forall i \in [L].$$

Lyapunov's indirect method of stability (Khalil, 2002; Sastry, 1999) states that given a dynamical system $x[k+1] = F(x[k])$, its equilibrium $x^*$ is stable in the sense of Lyapunov only if the linearization of the system around $x^*$

$$(x[k+1] - x^*) = (x[k] - x^*) + \left.\frac{\partial F}{\partial x}\right|_{x=x^*} (x[k] - x^*)$$

does not have any eigenvalue larger than 1 in magnitude. By using this fact for the system defined by (5)-(6), we can observe that an equilibrium $\{W_j^*\}_{j \in [L]}$ with $W_L^* \cdots W_1^* = \hat{R}$ is stable in the sense of Lyapunov only if the system

$$\left(E[k+1] - \hat{R} + R\right) = \left(E[k] - \hat{R} + R\right) - \delta \sum_{i=1}^{L} A_i\Big|_{\{W_j^*\}} \left(E[k] - \hat{R} + R\right) B_i\Big|_{\{W_j^*\}}$$

does not have any eigenvalue larger than 1 in magnitude, which requires that the mapping

$$f(\tilde{E}) = \sum_{i=1}^{L} A_i\Big|_{\{W_j^*\}} \tilde{E} B_i\Big|_{\{W_j^*\}} \tag{7}$$

does not have any real eigenvalue larger than $(2/\delta)$. Let $u$ and $v$ be the left and right singular vectors of $\hat{R}$ corresponding to its largest singular value, and let $p_j$ and $q_j$ be defined as in the statement of Theorem 1. Then, by Lemma 1, the mapping $f$ in (7) does not have an eigenvalue larger than $(2/\delta)$ only if

$$\sum_{i=1}^{L} p_{i-1}^2 q_{i+1}^2 \leq \frac{2}{\delta},$$

which completes the proof.                                                                     ∎

## Acknowledgement

This research was supported by the U.S. Office of Naval Research (ONR) MURI grant N00014-16-1-2710.

## Footnotes

[1]The random distribution must be continuous and assign zero probability to every set with measure zero.

[2]The code for the experiment is available at `https://github.com/nar-k/NeurIPS-2018`.

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
