[Supplementary Material]

# Supplementary Material

## A  Proof of Theorem 1 and Corollary 1

**Lemma 1.** *Let $A, B \in \mathbb{R}^{n \times n}$ be symmetric and positive semidefinite. Then, $\langle A, B \rangle \geq 0$.*

*Proof.* We can write $B$ as $B = \sum_{i=1}^{n} \lambda_i u_i u_i^\top$, where $\lambda_i \geq 0$ for all $i \in [n]$ and $u_i^\top u_j = 0$ if $i \neq j$. Then,

$$\langle A, B \rangle = \text{trace} \{AB\} = \text{trace} \left\{ A \sum_{i=1}^{n} \lambda_i u_i u_i^\top \right\} = \sum_{i=1}^{n} \lambda_i u_i^\top A u_i \geq 0. \qquad \blacksquare$$

**Lemma 2.** *Let $f : \mathbb{R}^{m \times n} \to \mathbb{R}^{m \times n}$ be a linear map defined as $f(X) = \sum_{i=1}^{L} A_i X B_i$, where $A_i \in \mathbb{R}^{m \times m}$ and $B_i \in \mathbb{R}^{n \times n}$ are symmetric positive semidefinite matrices for all $i \in [L]$. Then, for every nonzero $u \in \mathbb{R}^m$ and $v \in \mathbb{R}^n$, the largest eigenvalue of $f$ satisfies*

$$\lambda_{\max}(f) \geq \frac{1}{\|u\|_2^2 \|v\|_2^2} \sum_{i=1}^{L} (u^\top A_i u)(v^\top B_i v).$$

*Proof.* First, we show that $f$ is symmetric and positive semidefinite. Given two matrices $X, Y \in \mathbb{R}^{m \times n}$, we can write

$$\langle X, f(Y) \rangle = \text{trace} \left\{ \sum_i X^\top A_i Y B_i \right\} = \text{trace} \left\{ \sum_i B_i Y^\top A_i X \right\} = \langle Y, f(X) \rangle,$$

$$\langle X, f(X) \rangle = \text{trace} \left\{ \sum_i X^\top A_i X B_i \right\} = \sum_i \langle X^\top A_i X, B_i \rangle \geq 0,$$

where the last inequality follows from Lemma 1. This shows that $f$ is symmetric and positive semidefinite. Then, for every nonzero $X \in \mathbb{R}^{m \times n}$, we have

$$\lambda_{\max}(f) \geq \frac{1}{\langle X, X \rangle} \langle X, f(X) \rangle.$$

In particular, given two nonzero vectors $u \in \mathbb{R}^m$ and $v \in \mathbb{R}^n$,

$$\lambda_{\max}(f) \geq \frac{1}{\langle uv^\top, uv^\top \rangle} \langle uv^\top, f(uv^\top) \rangle = \frac{1}{\|u\|_2^2 \|v\|_2^2} \sum_{i=1}^{L} (u^\top A_i u)(v^\top B_i v). \qquad \blacksquare$$

**Proof of Theorem 1.** The cost function in Theorem 1 can be written as

$$\frac{1}{2} \text{trace} \left\{ (W_L \cdots W_1 - R)^\top (W_L \cdots W_1 - R) \right\}.$$

Let $E$ denote the error in the estimate, i.e. $E = W_L \cdots W_1 - R$. The gradient descent yields

$$W_i[k+1] = W_i[k] - \delta W_{i+1}^\top[k] \cdots W_L^\top[k] E[k] W_1^\top[k] \cdots W_{i-1}^\top[k] \quad \forall i \in [L]. \tag{1}$$

By multiplying the update equations of $W_i[k]$ and subtracting $R$, we can obtain the dynamics of $E$ as

$$E[k+1] = E[k] - \delta \sum_{i=1}^{L} A_i[k] E[k] B_i[k] + o(E[k]), \tag{2}$$

where $o(\cdot)$ denotes the higher order terms, and

$$A_i = W_L W_{L-1} \cdots W_{i+1} W_{i+1}^\top \cdots W_{L-1}^\top W_L^\top \quad \forall i \in [L],$$

$$B_i = W_1^\top W_2^\top \cdots W_{i-1}^\top W_{i-1} \cdots W_2 W_1 \quad \forall i \in [L].$$

Lyapunov's indirect method of stability (Khalil, 2002; Sastry, 1999) states that given a dynamical system $x[k+1] = F(x[k])$, its equilibrium $x^*$ is stable in the sense of Lyapunov only if the linearization of the system around $x^*$

$$(x[k+1] - x^*) = (x[k] - x^*) + \left. \frac{\partial F}{\partial x} \right|_{x = x^*} (x[k] - x^*)$$

does not have any eigenvalue larger than 1 in magnitude. By using this fact for the system defined by (1)-(2), we can observe that an equilibrium $\{W_j^*\}_{j\in[L]}$ with $W_L^* \cdots W_1^* = \hat{R}$ is stable in the sense of Lyapunov only if the system

$$\left(E[k+1] - \hat{R} + R\right) = \left(E[k] - \hat{R} + R\right) - \delta \sum_{i=1}^{L} A_i\Big|_{\{W_j^*\}} \left(E[k] - \hat{R} + R\right) B_i\Big|_{\{W_j^*\}}$$

does not have any eigenvalue larger than 1 in magnitude, which requires that the mapping

$$f(\tilde{E}) = \sum_{i=1}^{L} A_i\Big|_{\{W_j^*\}} \tilde{E} B_i\Big|_{\{W_j^*\}} \tag{3}$$

does not have any real eigenvalue larger than $(2/\delta)$. Let $u$ and $v$ be the left and right singular vectors of $\hat{R}$ corresponding to its largest singular value, and let $p_j$ and $q_j$ be defined as in the statement of Theorem 1. Then, by Lemma 2, the mapping $f$ in (3) does not have an eigenvalue larger than $(2/\delta)$ only if

$$\sum_{i=1}^{L} p_{i-1}^2 q_{i+1}^2 \leq \frac{2}{\delta},$$

which completes the proof. ∎

**Proof of Corollary 1.** Note that

$$q_{i+1} p_i = \|u^\top W_L W_{L-1} \cdots W_{i+1}\|_2 \|W_i \cdots W_2 W_1 v\|_2 \geq \|u^\top W_L \cdots W_1 v\|_2 = \rho(R).$$

As long as $\rho(R) \neq 0$, we have $p_i \neq 0$ for all $i \in [L]$, and therefore,

$$p_{i-1}^2 q_{i+1}^2 \geq \frac{p_{i-1}^2}{p_i^2} \rho(R)^2. \tag{4}$$

Using inequality (4), the bound in Theorem 1 can be relaxed as

$$\delta \leq 2 \left(\sum_{i=1}^{L} \frac{p_{i-1}^2}{p_i^2} \rho(R)^2\right)^{-1}. \tag{5}$$

Since $\prod_{i=1}^{L}(p_i/p_{i-1}) = \rho(R) \neq 0$, we also have the inequality

$$\sum_{i=1}^{L} \frac{p_{i-1}^2}{p_i^2} \rho(R)^2 \geq \sum_{i=1}^{L} \frac{\rho(R)^2}{\left(\rho(R)^{1/L}\right)^2} = L\rho(R)^{2(L-1)/L},$$

and the bound in (5) can be simplified as

$$\delta \leq \frac{2}{L\rho(R)^{2(L-1)/L}}. \qquad ∎$$

## B  Proof of Theorem 2

**Lemma 3.** *Let $\lambda > 0$ be estimated as a multiplication of the scalar parameters $\{w_i\}_{i\in[L]}$ by minimizing $\frac{1}{2}(w_L \cdots w_2 w_1 - \lambda)^2$ via gradient descent. Assume that $w_i[0] = 1$ for all $i \in [L]$. If the step size $\delta$ is chosen to be less than or equal to*

$$\delta_c = \begin{cases} L^{-1}\lambda^{-2(L-1)/L} & \text{if } \lambda \in [1, \infty), \\ (1-\lambda)^{-1}(1-\lambda^{1/L}) & \text{if } \lambda \in (0, 1), \end{cases}$$

*then $|w_i[k] - \lambda^{\frac{1}{L}}| \leq \beta(\delta)^k |1 - \lambda^{\frac{1}{L}}|$ for all $i \in [L]$, where*

$$\beta(\delta) = \begin{cases} 1 - \delta(\lambda - 1)(\lambda^{1/L} - 1)^{-1} & \text{if } \lambda \in (1, \infty), \\ 1 - \delta L\lambda^{2(L-1)/L} & \text{if } \lambda \in (0, 1]. \end{cases}$$

*Proof.* Due to symmetry, $w_i[k] = w_j[k]$ for all $k \in \mathbb{N}$ for all $i, j \in [L]$. Denoting any of them by $w[k]$, we have

$$w[k+1] = w[k] - \delta w^{L-1}[k](w^L[k] - \lambda).$$

To show that $w[k]$ converges to $\lambda^{1/L}$, we can write

$$w[k+1] - \lambda^{1/L} = \mu(w[k])(w[k] - \lambda^{1/L}),$$

where

$$\mu(w) = 1 - \delta w^{L-1} \sum_{j=0}^{L-1} w^j \lambda^{(L-1-j)/L}.$$

If there exists some $\beta \in [0, 1)$ such that

$$0 \le \mu(w[k]) \le \beta \text{ for all } k \in \mathbb{N}, \tag{6}$$

then $w[k]$ is always larger or always smaller than $\lambda^{1/L}$, and its distance to $\lambda^{1/L}$ decreases by a factor of $\beta$ at each step. Since $\mu(w)$ is a monotonic function in $w$, the condition (6) holds for all $k$ if it holds only for $w[0] = 1$ and $\lambda^{1/L}$, which gives us $\delta_c$ and $\beta(\delta)$. ∎

**Proof of Theorem 2.** There exists a common invertible matrix $U \in \mathbb{R}^{n \times n}$ that can diagonalize all the matrices in the system created by the gradient descent: $R = U \Lambda_R U^\top$, $W_i = U \Lambda_{W_i} U^\top$ for all $i \in [L]$. Then the dynamical system turns into $n$ independent update rules for the diagonal elements of $\Lambda_R$ and $\{\Lambda_{W_i}\}_{i \in [L]}$. Lemma 3 can be applied to each of the $n$ systems involving the diagonal elements. Since $\delta_c$ in Lemma 3 is monotonically decreasing in $\lambda$, the bound for the maximum eigenvalue of $R$ guarantees linear convergence. ∎

## C   Proof of Theorem 3

**Lemma 4.** *Assume that $\lambda < 0$ and $w_i[0] = 1$ is used for all $i \in [L]$ to initialize the gradient descent algorithm to solve*

$$\min_{(w_1, \dots, w_L) \in \mathbb{R}^L} \frac{1}{2} \left( w_L \dots w_2 w_1 - \lambda \right)^2.$$

*Then, each $w_i$ converges to 0 unless $\delta > (1 - \lambda)^{-1}$.*

*Proof.* We can write the update rule for any weight $w_i$ as

$$w[k+1] = w[k] \left( 1 - \delta \sigma w^{L-2}[k] \left( w^L[k] - \lambda \right) \right)$$

which has one equilibrium at $w^* = \lambda^{1/L}$ and another at $w^* = 0$. If $0 < \delta \le 1/\sigma(1 - \lambda)$ and $w[0] = 1$, it can be shown by induction that

$$0 \le 1 - \delta \sigma w^{L-2}[k] \left( w^L[k] - \lambda \right) < 1$$

for all $k \ge 0$. As a result, $w[k]$ converges to 0. ∎

**Proof of Theorem 3.** Similar to the proof of Theorem 2, the system created by the gradient descent can be decomposed into $n$ independent systems of the diagonal elements of the matrices $\Lambda_R$ and $\{\Lambda_{W_i}\}_{i \in [L]}$. Then, Lemma 3 and Lemma 4 can be applied to the systems with positive and negative eigenvalues of $R$, respectively. ∎

## D   Proof of Theorem 4

To find a necessary condition for the convergence of the gradient descent algorithm to $(\hat{W}, \hat{V})$, we analyze the local stability of that solution in the sense of Lyapunov. Since the analysis is local and the function $g$ is fixed, for each point $x_i$ we can use a matrix $G_i$ that satisfies $G_i(\hat{V}x_i - b) = g(\hat{V}x_i - b)$. Note that $G_i$ is a diagonal matrix and all of its diagonal elements are either 0 or 1. Then, we can write the cost function around an equilibrium as

$$\frac{1}{2} \sum_{i=1}^{N} \text{trace} \left\{ [W G_i(V x_i - b) - f(x_i)]^\top [W G_i(V x_i - b) - f(x_i)] \right\}.$$

Denoting the error $W G_i(V x_i - b) - f(x_i)$ by $e_i$, the gradient descent gives

$$W[k+1] = W[k] - \delta \sum_{i=1}^{N} e_i[k](V[k]x_i - b)^\top G_i^T,$$

$$V[k+1] = V[k] - \delta \sum\nolimits_{i=1}^{N} G_i^\top W[k]^\top e_i[k] x_i^\top.$$

Let $e$ denote the vector $(e_1^\top \ \ldots \ e_N^\top)^\top$. Then we can write the update equation of $e_j$ as

$$\begin{aligned} e_j[k+1] &= e_j[k] - \delta W[k] G_j \sum\nolimits_i G_i^\top W[k]^\top e_i[k] x_i^\top x_j \\ &\quad - \delta \sum\nolimits_i e_i[k] (V[k]x_i - b)^\top G_i^\top G_j (V[k]x_j - b) + o(e[k]). \end{aligned}$$

Similar to the proof of Theorem 1, the equilibrium $(\hat{W}, \hat{V})$ can be stable in the sense on Lyapunov only if the system

$$e_j[k+1] = e_j[k] - \delta \sum\nolimits_i \hat{W} G_j G_i^\top \hat{W}^\top e_i[k] x_i^\top x_j - \delta \sum\nolimits_i e_i[k] (\hat{V}x_i - b)^\top G_i^\top G_j (\hat{V}x_j - b) \quad (7)$$

does not have any eigenvalue larger than 1 in magnitude. Note that the linear system in (7) can be described by a symmetric matrix, whose eigenvalues cannot be larger in magnitude than the eigenvalues of its sub-blocks on the diagonal, in particular those of the system

$$e_j[k+1] = e_j[k] - \delta \hat{W} G_j G_j^\top \hat{W}^\top e_j[k] x_j^\top x_j - \delta e_j[k] (\hat{V}x_j - b)^\top G_j^\top G_j (\hat{V}x_j - b). \quad (8)$$

The eigenvalues of the system (8) are less than 1 in magnitude only if the eigenvalues of the system

$$h(u) = \hat{W} G_j G_j^\top \hat{W}^\top u x_j^\top x_j + u(\hat{V}x_j - b)^\top G_j^\top G_j (\hat{V}x_j - b)$$

are less than $(2/\delta)$. This requires that for all $j \in [N]$ for which $\hat{f}(x_j) \neq 0$,

$$\begin{aligned} \frac{2}{\delta} &\geq \frac{\langle \hat{f}(x_j), h(\hat{f}(x_j)) \rangle}{\langle \hat{f}(x_j), \hat{f}(x_j) \rangle} \\ &= \frac{1}{\|\hat{f}(x_j)\|^2} \left( \|G_j^\top \hat{W}^\top \hat{f}(x_j)\|^2 \|x_j\|^2 + \|\hat{f}(x_j)\|^2 \|G_j(\hat{V}x_j - b)\|^2 \right) \\ &\geq \frac{1}{\|\hat{f}(x_j)\|^2} \frac{\|(\hat{V}x_j - b)^\top G_j^\top G_j^\top \hat{W}^\top \hat{f}(x_j)\|^2}{\|(\hat{V}x_j - b)^\top G_j^\top\|^2} \|x_j\|^2 + \|G_j(\hat{V}x_j - b)\|^2 \\ &= \frac{1}{\|G_j(\hat{V}x_j - b)\|^2} \|\hat{f}(x_j)\|^2 \|x_j\|^2 + \|G_j(\hat{V}x_j - b)\|^2 \\ &\geq 2\|\hat{f}(x_j)\| \|x_j\|. \end{aligned}$$

As a result, Lyapunov stability of the solution $(\hat{W}, \hat{V})$ requires

$$\frac{1}{\delta} \geq \max_i \|\hat{f}(x_i)\| \|x_i\|. \qquad\qquad \blacksquare$$