[Reviews · NeurIPS 2018]

Reviewer 1



This paper proves several results related to the step size (learning rate) in gradient Descent (GD). First, the paper examines linear feedforward neural nets with a quadratic loss, trained on whitened data with outputs which are a PSD matrix times the input. It is found that GD can only converge to a critical point if the learning rate it is below a certain threshold. Then, it is proven that the weights converge at a linear rate to the global minimum if they are initialized to identity, and the step size is below another threshold. The latter result is also extended to matrices with negative eigenvalues. Second, the paper examines a single hidden layer neural network with quadric loss, and finds that the weights can converge to a critical point only if the learning rate is below another ratio. Clarity: The paper is easily readable. Quality: I couldn't find major problems (some minor points below). Significance: These results suggest another explanation for why deeper networks are more challenging to train (since we require a lower step size for convergence), why residual connections are helpful (easier convergence to the global minima), and why the training algorithm (GD) is biased towards global minima with smaller "sharpness" (higher sharpness makes global minima more unstable) and possibly better generalization . Originality: I'm surprised this analysis hasn't been done before. I think it is common knowledge that on smooth functions we require 2/(step size) be larger than the max eigenvalue of the Hessian for GD to converge for almost every initialization (in a related subject, if 2/(step size) is larger than the global smoothness, then this is sufficient for convergence, e.g., https://rkganti.wordpress.com/2015/08/20/unconstrained-minimization-gradient-descent-algorithm/). However, the analysis following from this fact (as well as the other results) seems novel to me. Minor points: 1) Lyapunov instability implies that we can't to converge to fixed point *independently of initialization*, not in general (since for certain initializations, on stable manifolds, we may still converge). The authors mention this in certain places in the text, but the phrasing in the Theorems missing this distinction (e.g. in Theorem 1 we only have "the gradient descent algorithm can converge to a solution" without mention of the initialization). This makes the Theorems 1 and 4 (and Corollary 2) technically wrong, so it should be corrected. 2) line 209: "provides a case where the bound (4) is tight" seems inaccurate: note the bound in eq 4 is x2 the bound in Theorem 2 (in the right case of the max), so it's nearly tight. 3) In the numerical results part, I would mention that the bound from Theorem 4 gives \delta<3.8e-4, which is close to the range of values you found numerically (in which the destabilization occurs). So while the bound is not tight, it seems to be in the right ballpark. 4) I recommend the authors to make their code available online. %% Edited after author feedback %% Thank you for correcting these issue. One last thing I noticed - we can replace the identity initialization with a diagonal initialization, and everything will hold, right? Maybe worth mentioning what is the largest family of initializations that will work, as this can be of practical intreset (e.g., I think that in RNNs diagonal initializations sometimes work better than identity initialization).

Reviewer 2



This paper analyzes the effect of the choice of step length for the (full-batch) gradient method applied to feed-forward linear networks and two-layer (one hidden layer) ReLU networks from a discrete-time dynamical system perspective. In particular, the bounds on the steplength characterize which minima the algorithm can converge to when certain steplengths are used in the algorithm. A more in-depth analysis is also provided for the case of symmetric positive definite matrices, in which the maximum steplength can be used when the parameters are initialized with the identity matrix. Overall, the paper provides novel bounds on the step length necessary to converge to a certain subset of stationary points. This corresponds with my understanding that practitioners are indeed selecting their initialization and learning rate schedules for SGD to specifically converge to particular local minima that have better generalization properties. Although these theoretical results are very interesting, they are motivated improperly, in my opinion. I expand on this point below: 1. Explanation of Training Error Behavior Coming from a background in optimization rather than dynamical systems, the first contribution (line 80) highlighting the reason the training error stops decreasing is already well understood from an optimization perspective, in my opinion. In particular, the convergence of the gradient method for strongly convex functions with fixed step length requires Lipschitz continuity of the gradient (or smoothness). This makes some claims (such as the claim on large singular values in line 263-264) trivial. If the smoothness condition does not hold, one may have to diminish the step length in order to converge to the solution. In the more realistic case in which we are applying the stochastic (sub)gradient method, we know that it indeed requires a diminishing steplength (typically non-summable but square-summable) in order to converge to the solution; otherwise, the algorithm only is guaranteed to converge to a neighborhood of the solution. See Bottou, et al. (2018) for a review on the convergence analysis of stochastic gradient methods. In this sense, the claim that the (stochastic) gradient method is converging to a periodic orbit in dynamical systems is similar to the results of only having guarantees to converge to a neighborhood of the solution, which is already sufficient to explain the plots from He, et al. (2016). Rather than motivating the theoretical results from this angle (which I would consider to be uninteresting and well-known from optimization), I would suggest the authors take the approach of claiming to provide theory that better elucidates the connection between the choice of step length and the subset of local optima the algorithm can converge to, from a dynamical systems perspective. This is the part of the paper that I would consider most novel and interesting, specifically towards understanding the training of neural networks. 2. Stochastic vs. Deterministic Unfortunately, the paper fails to address how the use of stochasticity in the stochastic gradient method could impact these results for the choice of step length. Instead, the paper only analyzes the case for the full gradient method, which is not representative of what is used in practice (which also takes away from the first contribution). If these results claim to establish the relationship between the choice of step length for training neural networks, the paper must address the lack of stochasticity of the method considered in the analysis and use these results as a theoretical building block towards understanding the stochastic gradient method for training neural networks. 3. Relationship with Lipschitz Continuity and Smoothness Lastly, I would suggest the authors provide a more in-depth investigation of the connection between these results and the local Lipschitz and smoothness constants of the objective function. I’m curious how these results may be related to the typical convergence analysis of the gradient method in optimization. For instance, in the first example (line 36), the function considered does not have a global smoothness constant; hence, the gradient method would require a diminishing step length in order to converge from any initialization. This is clearly reflected in the example. Minor Comments and Questions: - Should always say “the gradient method” or “the gradient descent algorithm”, not “the gradient descent”. - The networks considered are not relevant to practice, but this is OK given that the analysis of neural networks have mainly been limited to linear and ReLU feed-forward networks. - The paper is well-written. Summary: In summary, this paper provides some elucidating theoretical results on the set of local minima that the gradient method will converge to for certain choices of step size for simple feed-forward networks. This could be motivated better from a different angle as described earlier, without the comment explaining the convergence of the method to a periodic orbit or neighborhood of the solution, which is well-known in optimization. Because of its theoretical novelty, I believe that this well-written paper is indeed worthy of publication, with some minor edits and perhaps a major change in approach and introduction. ================================ Post-Author Response: I thank the authors for their well-crafted response, particularly for clarifying the comment made in Lines 263-264. In light of the changes proposed in the response regarding the introduction and additional remarks, I have raised my score to a 7.

Reviewer 3



The paper provides an interesting analysis of gradient descent for deep linear networks as well as ReLU networks with only one hidden layer. Both cases use squared loss. The paper is well-written and well-organized and builds on tools from discrete-time dynamical systems. The paper has several interesting (and to the best of my knowledge original) components: 1. Providing upper bounds on step size for reaching each solution of a linear network. 2. For learning positive definite matrices by linear networks, identity initialization of weights allows the largest step size. 3. Analysis of the solution obtained by linear networks initialized by identity weights when the target matrix has negative eigenvalues. 4. Relationship between step size and model's output for a single hidden layer network with ReLU activations. Question: Q1. Figure 2 shows with the smaller learning rate, the estimated function converges to a solution, and with the larger one it oscillates. I am wondering if the convergence/oscillation claim here is made based on mathematically analysis the resulted dynamical system or is judged based on numerically computed notion residual? If the latter, couldn't it be the case that the oscillation are not really perfect and are slowly converging to the target function? Since your point here is not speed of convergence, but rather realization of a perfect convergence or oscillation, numerical assessment may be misleading if that is what you are doing. Q2. The paper has a general result for deep linear networks to learn an arbitrary nonzero matrix (Theorem 1 and Corollary 1). Specifically, in Corollary 1 authors provide an upper bound on the learning rate based on the largest singular value of the matrix. Later in the paper, when they add further assumption that the matrix is positive definite and weight matrices are initialized by identity, the step size bound becomes tighter. I am wondering if the authors have any intuition about what a worst case matrix and initialization would be for the general linear network case, to get a better understanding of the quality of the general-case bound. Minor comment: In Example 1, the specification of the set S as S={0,d^2,-d^2,...} is ambiguous. Adding one more pair before "..." would help. POST REBUTTAL: I had some minor questions which the authors replied. My rating for the paper is 8.